# Peer review of "Retrospective Study on the Clinical Superiority of the Vacuum-Assisted Closure System with a Silicon-Based Dressing over the Conventional Tie-over Bolster Technique in Skin Graft Fixation"

_medicina, 2019, doi:10.3390/medicina55120781_

Round 1

Reviewer 1 Report

The article is interesting and well written. It was reported the use of vacuum-assisted closure (VAC) with a silicon-based dressing as an alternative for skin graft fixation.

The Discussion section must be improved a lot, reporting the results published in literature on the use of VAC therapy in wound healing (also loss of substance, chronic and post traumatic ulcers etc) comparing with other biotechnologies as Platelet rich plasma (PRP), Adipose derived stem cells (ASCs) or Stromal Vascular Fraction Cells (SVFs) contained in fat graft, Dermal substitute, Hyaluronic acid etc. also ranging in other fields of wound healing, such as ulcers, loss of substance, soft tissue defects (outcomes of reconstruction, outcomes of scars, outcomes of burns etc) bone tissue defects, highlighting the potential use of VAC and these biotechnologies in a lot of kinds wound.

In addition, the authors’ goals must be to review a major number interesting paper a cellular mechanism approach in order to regenerate damaged tissue and promote one’s own natural GFs release.

On the other hand, it is necessary to compare these results with old and traditional studies suggest the use of a bone-anchored implant as camouflage or implants (prostheesis) or autologous graft (skin but also cartilage and bone.

Author Response

Dear Reviewer:

Thank you very much for your comments on our manuscript. We have carefully considered the comments of the reviewers and revised our manuscript accordingly. The revised parts have been highlighted in the manuscript. Our responses to the reviewers' comments are as follows.

Major Criticisms:

The Discussion section must be improved a lot, reporting the results published in literature on the use of VAC therapy in wound healing (also loss of substance, chronic and post traumatic ulcers etc) comparing with other biotechnologies as Platelet rich plasma (PRP), Adipose derived stem cells (ASCs) or Stromal Vascular Fraction Cells (SVFs) contained in fat graft, Dermal substitute, Hyaluronic acid etc. also ranging in other fields of wound healing, such as ulcers, loss of substance, soft tissue defects (outcomes of reconstruction, outcomes of scars, outcomes of burns etc) bone tissue defects, highlighting the potential use of VAC and these biotechnologies in a lot of kinds wound.

 ANSWER: 

  Thank you for Reviewer’s concern, we agree with your points. Advanced biotechnologies such as PRP, ASCs, SVFs, dermal substitute, and Hyaluronic acid are great examples for applying with VAC in various types of trauma in lower extremities. We will take these new combinations of biotechnologies into clinical consideration and evaluate its efficacy for better postoperative wound care management.

In addition, the authors’ goals must be to review a major number interesting paper a cellular mechanism approach in order to regenerate damaged tissue and promote one’s own natural GFs release.

ANSWER:

  Thank you for Reviewer’s suggestion, we agree with your points. Regenerative medicine has now been well-developed and promising in Modern medicine, especially in the field of “tissue engineering”. We will do further research based on basic medicine to study the underlying regenerative cellular mechanism (the release of GF, VEGF, etc.) of the combination of VAC and silicon-based dressing.

On the other hand, it is necessary to compare these results with old and traditional studies suggest the use of a bone-anchored implant as camouflage or implants (prosthesis) or autologous graft (skin but also cartilage and bone.

ANSWER:

  Thank you for Reviewer’s comment, we agree with your points. Traditional studies on bone-anchored implant as camouflage or implants (prostheesis) or autologous graft (skin but also cartilage and bone are great example and target for further comparison with the combination of VAC and silicon-based dressing. It is an incredible suggestion for us to evaluate traditional wound care management other than tie-over bolster technique in future studies.

Reviewer 2 Report

Aim:

it is unclear what was the aim of the study - to research the pain level between groups, skin graft take rates or postoperative complications? This has to be very clearly stated in abstract, introduction, proven through statistics and stated in conclusion.

Line 79 - this sentence misleads the reader, as 60 patients were approached, and 26 of them were included in the study.

Methods:

You mentioned lots of exclusion criteria, I advise inserting a flowchart of patient selection. Please explain wound etiology, can it have an impact on research results? Explain why you have chosen this kind of mesh and is it similar and how to Mepitel which is mentioned in the discussion section. Explain the type of negative pressure device, you mentioned only foam type.

Discussion:

Reference 22 is not well connected with the research subject, so exclude it. You rate postoperative pain, but in the discussion cited is literature which discusses usage of silicone dressing and its effect on pain at wound dressing.

Conclusion

Same comment as for abstract - you have to conclude what you mentioned for aims, as the conclusion is based on the results, I suggest to change the aims in the abstract.

Also, I'll prefer term negative wound pressure therapy, as VAC reminds on the protected name, if so, please clarify it.

Author Response

Dear Reviewer:

Thank you very much for your comments on our manuscript. We have carefully considered the comments of the reviewers and revised our manuscript accordingly. The revised parts have been highlighted in the manuscript. Our responses to the reviewers' comments are as follows.

Major Criticisms:

it is unclear what was the aim of the study - to research the pain level between groups, skin graft take rates or postoperative complications? This has to be very clearly stated in abstract, introduction, proven through statistics and stated in conclusion. Same comment as for abstract - you have to conclude what you mentioned for aims, as the conclusion is based on the results, I suggest to change the aims in the abstract.

ANSWER: 

  Thank you for Reviewer’s suggestion, we agree with your points and we had described this in abstract part (page 1, paragraph line 24-26), and conclusions part (page 12, paragraph line 338-340).

Line 79 - this sentence misleads the reader, as 60 patients were approached, and 26 of them were included in the study.

ANSWER:

    Thank you for Reviewer’s suggestion, we agree with your points and we had revised this in methods part (page 2, paragraph line 80-82) for clearer understanding.

You mentioned lots of exclusion criteria, I advise inserting a flowchart of patient selection.

ANSWER:   

    Thank you for Reviewer’s advice, we agree with your points and we had added flowchart of patient selection in figure 1.

Please explain wound etiology, can it have an impact on research results?

ANSWER:

  Thank you for Reviewer’s question and concern, we agree with your points and we had revised it in results part (page 7, paragraph 165-167). Wound etiology of our patients included various type of trauma on lower extremities which require full-thickness skin graft, such as necrotizing fasciitis, surgical defect of skin cancer, burn, dog bite, and diabetic wound. In this study, the aim is to evaluate difference in performance between 2 wound care managements, thus we didn’t take the impact of wound etiology into our main consideration. However, your question is a perfect spark for an idea for further study on whether the performance of wound care management would be affected by different wound etiology.

Explain why you have chosen this kind of mesh and is it similar and how to Mepitel which is mentioned in the discussion section.

ANSWER:

  In fact, the properties of the silicon-based dressing used in this study are similar to Mepitel in essence. Although there are applications of Mepitel-like dressing with VAC therapy as mentioned by references in discussion part, there are lack of comparison study on clinical efficacy between Mepitel and silicon-based dressing used in this study. For the purpose of trying a new combination in postoperative wound are management for skin graft surgery, we therefore chosen silicon-based dressing (SI-mesh; ALCARE Tokyo, Japan) to apply with VAC therapy.

Explain the type of negative pressure device, you mentioned only foam type.

ANSWER:   

    Thank you for Reviewer’s comment, we agree with your points and we had revised it in method part, 2.2.2. VAC therapy with silicon-based dressing (page 4, paragraph 124).

Reference 22 is not well connected with the research subject, so exclude it. You rate postoperative pain, but in the discussion cited is literature which discusses usage of silicone dressing and its effect on pain at wound dressing.

ANSWER:

  Thanks for your precious remind and suggestion. However, after checking, reference 22 is about longer operative time, higher rate of postoperative complications, which is actually match to the section in discussion part on the issue of operative time.

Also, I'll prefer term negative wound pressure therapy, as VAC reminds on the protected name, if so, please clarify it.

ANSWER:

  Thanks for your suggestion, we had described more in introduction part (page2, paragraph 58-60).

Round 2

Reviewer 2 Report

Dear authors,

thank you very much for your accurate and fast answers, however, there are still some issues that need correction.

In the previous comment, I suggested you to clearly state what is the aim of the study, please clearly indicate it, the reader doesn't know it.

In the exclusion criteria flowchart, please indicate how many patients and because of what etiology were excluded.

Wound etiology - yes, it is a problem because wound etiology can have a great impact on wound healing; try to solve this - maybe you had all acute wounds or you can otherwise explain that wound etiology didn't have an impact on graft surviving.

Author Response

Dear Reviewer:

Thank you very much for your comments on our manuscript. We have carefully considered the comments of the reviewers and revised our manuscript accordingly. The revised parts have been highlighted with track changes in the manuscript. Our responses to the reviewers' comments are as follows.

Major Criticisms:

In the previous comment, I suggested you to clearly state what is the aim of the study, please clearly indicate it, the reader doesn't know it.

ANSWER: 

  Thank you for Reviewer’s suggestion, we agree with your points and we had further described this in abstract part (page 1, paragraph line 24-27), and conclusions part (page 12, paragraph line 345-347).

In the exclusion criteria flowchart, please indicate how many patients and because of what etiology were excluded.

ANSWER:

    Thank you for Reviewer’s remind, we agree with your points and we had reinvestigated our data and revised this in the flowchart (Figure 1).

Wound etiology - yes, it is a problem because wound etiology can have a great impact on wound healing; try to solve this - maybe you had all acute wounds or you can otherwise explain that wound etiology didn't have an impact on graft surviving.

ANSWER:   

    Thank you for Reviewer’s advice, we agree with your points and we had added further description in results part (page 7, paragraph 172-173). Every wound bed in this study was well-prepared to ensure no impact of wound etiology on clinical outcome.